# H²CL: Heterogeneity-Aware Hypergraph Contrastive Learning for Robust Representation Learning

**Kaixuan Yao** [1]   **Ting Guo** [2]   **Feilong Cao** [3]   **Ming Li** [4]

## Abstract

Hypergraph contrastive learning has shown strong potential for modeling high-order relations, but most existing methods implicitly assume that nodes within the same hyperedge are semantically homogeneous. In real-world hypergraphs, this assumption often breaks down: heterogeneous node–hyperedge interactions may introduce noisy signals during augmentation and message passing, leading to feature contamination and false-positive contrastive alignment. To address this issue, we propose H²CL, a heterogeneity-aware hypergraph contrastive learning framework for robust representation learning. H²CL first estimates node–hyperedge heterogeneity from input features and uses it to guide a Heterogeneity-Aware View Generator, which selectively masks high-heterogeneity incidences and constructs cleaner contrastive views. It further introduces a Heterogeneity-Aware Hypergraph Encoder that dynamically reweights information propagation in both node-to-hyperedge and hyperedge-to-node aggregation, enabling hyperedges to aggregate more homogeneous signals while suppressing heterogeneous noise. We also provide theoretical analysis showing that the encoder corresponds to a coordinate descent step for minimizing a heterogeneity-weighted Dirichlet energy. Extensive experiments on standard benchmarks and larger-scale hypergraphs demonstrate that H²CL achieves competitive or superior

performance compared with recent baselines, remains robust under structural noise, and learns reweighting patterns that effectively reduce heterogeneity during message passing. Our code is available at: https://github.com/sxu-yaokx/HHCL

## 1. Introduction

With the advent of the information age, fields such as social networks (Han et al., 2022), knowledge graphs (Xia et al., 2022b), and recommendation systems (Xia et al., 2022c) have accumulated massive amounts of graph-structured data. These data encompass multidimensional complex relationships among users, content, and interactions. Extracting high-quality feature representations from such graph-structured data has become a critical research challenge in the data engineering domain. As a generalization of graphs, hypergraphs not only represent pairwise relationships of ordinary graphs but also inherently describe multi-to-multi relationships between nodes and edges through hyperedges (Song et al., 2024; Xia et al., 2022c). This structural flexibility provides natural advantages in capturing grouped relationships, enabling finer semantic and structural representations (Yadati et al., 2019; Zhang et al., 2021).

However, a critical limitation of traditional hypergraph learning paradigms is their implicit assumption of *homophily*—that nodes sharing a hyperedge share similar semantic features or labels. In complex real-world scenarios, this assumption breaks down, giving rise to the *hypergraph heterogeneity phenomenon*. Specifically, a single hyperedge encapsulates nodes from diverse semantic categories or distributions due to the multifaceted nature of high-order interactions (e.g., interdisciplinary collaborations). Standard message passing mechanisms, which aggregate neighborhood information indiscriminately, fail to distinguish these semantic variations. Consequently, in homophily-dependent node classification and semantic representation learning, heterogeneous nodes can act as sources of "noise," propagating irrelevant information during aggregation. This leads to *feature contamination*, where the distinctiveness of node representations is diluted by inconsistent signals from their neighbors. The relevance of this issue is task-dependent:

[1]School of Computer and Information Technology, the Key Laboratory of Computational Intelligence and Chinese Information Processing of Ministry of Education, Shanxi University, Taiyuan, China [2] School of Computer Science and Technology, North University of China, Taiyuan, China [3]School of Mathematics, Institute of Mathematics and Cross-disciplinary Science, Zhejiang Normal University, China [4]Zhejiang Key Laboratory of Intelligent Education Technology and Application, Zhejiang Normal University, Jinhua, China. Correspondence to: Ming Li <mingli@zjnu.edu.cn>.

*Proceedings of the 43ʳᵈ International Conference on Machine Learning*, Seoul, South Korea. PMLR 306, 2026. Copyright 2026 by the author(s).

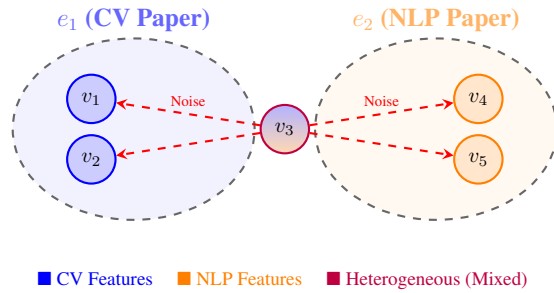

*Figure 1.* Illustration of the Hypergraph Heterogeneity Phenomenon.

bridge nodes may be useful signals in applications such as influence propagation, while our focus is robust class-separable representation learning. As illustrated in Figure 1, consider a co-authorship hypergraph where hyperedges represent papers. A researcher ($v_3$) working at the intersection of Computer Vision (CV) and Natural Language Processing (NLP) may bridge two distinct communities. While structurally connected to both, $v_3$ introduces semantic heterogeneity: aggregating the features of such bridge nodes indiscriminately can "pollute" the specialized representations of pure CV or NLP researchers ($v_1, v_2, v_4, v_5$), thereby degrading the model's discriminative power.

While Contrastive Learning (CL) offers a promising avenue to mitigate representation bias by maximizing cross-view consistency (Xia et al., 2022a; Qiu et al., 2020), current hypergraph CL paradigms remain suboptimal. The fundamental flaw lies in their reliance on *randomized* augmentation strategies (e.g., uniform node masking or hyperedge perturbation). Such topology-agnostic approaches fail to discriminate between homogeneous and heterogeneous nodes. Consequently, they risk preserving or even amplifying the feature contamination identified above, as the "noisy" bridge nodes are just as likely to be retained as the core semantic nodes. This inability to explicitly filter heterogeneous noise during view generation limits the model's ability to learn robust, noise-invariant representations.

To overcome these limitations, we propose **H²CL**, a novel **H**eterogeneity-aware **H**ypergraph **C**ontrastive **L**earning framework. Our method fundamentally redefines the role of hyperedges: rather than passive channels for information flow, we utilize them as active *semantic filters* to purify node representations. This is achieved through two coupled mechanisms: (1) A *Heterogeneity-Aware View Generator* that replaces random masking with a targeted suppression strategy, preferentially masking high-heterogeneity incidences to construct "cleaner" contrastive views; and (2) A *Heterogeneity-Aware Hypergraph Encoder* that dynamically modulates message passing weights. By penalizing information flow from heterogeneous connections during both the node-to-hyperedge and hyperedge-to-node phases,

our encoder effectively aggregates homogeneous signals while blocking noise. Furthermore, we provide theoretical guarantees showing that this mechanism minimizes a heterogeneity-weighted Dirichlet energy, formally proving its capability to enforce semantic smoothness. The main contributions of this work are summarized as follows:

- We propose H²CL, a heterogeneity-aware hypergraph contrastive learning framework. Unlike existing methods that rely on passive aggregation, H²CL introduces an active semantic filtering paradigm to mitigate feature contamination, offering a robust solution for representation learning in high-heterogeneity networks.

- We provide a rigorous theoretical analysis proving that our Heterogeneity-Aware Encoder functions as a coordinate descent step that minimizes a specific Heterogeneity-Weighted Dirichlet Energy. This mathematically guarantees that the model enforces feature smoothness strictly within homogeneous clusters while sharpening boundaries at heterogeneous interfaces.

- We demonstrate that our Heterogeneity-Aware View Generator mitigates the "False Positive" problem inherent in random masking. Under a heterogeneity-semantic calibration assumption, the generated views are better aligned with the clean semantic incidence distribution than uniform random masking, so the contrastive objective is less affected by structural noise.

## 2. Related Work

### 2.1. Hypergraph Neural Networks

In recent years, hypergraph structures have attracted widespread attention due to their powerful high-order information representation capabilities. Hypergraph Neural Networks (HGNN) (Feng et al., 2019) have unique advantages in modeling multiple relationships, and have performed well on complex data structures (such as social networks, knowledge graphs, and biological information, etc.). UniGNN (Huang & Yang, 2021) has explored the information propagation of traditional graphs and hypergraphs, extending the general message-passing process of GNNs to hypergraphs, but it has not proposed a more suitable message passing method for hypergraph structures. However, these methods are all based on the assumption of homogeneous neighborhood information for aggregating neighborhood information, and do not effectively handle the impact brought by heterogeneous information. Attention-based hypergraph encoders such as HyperGAT (Ding et al., 2020) learn node-hyperedge importance weights implicitly in supervised tasks, whereas H²CL explicitly estimates heterogeneity and uses it in both self-supervised view construction and two-stage propagation.

## 2.2. Graph and Hypergraph Contrastive Learning

Contrastive Learning (CL) is a self-supervised learning method, which learns by minimizing the distance between similar samples and maximizing the distance between different samples. Graph Contrastive Learning (GCL) applies the idea of CL to Graph Neural Networks. GCL (Xia et al., 2022a; Qiu et al., 2020), as a self-supervised or unsupervised learning method, enables the model to learn high-quality feature representations without labels by contrasting embeddings in different views. Although the GCL model performs well in graph representation learning, it is not suitable for hypergraphs for the following two reasons. Firstly, applying the GCL model to hypergraphs requires converting hypergraphs to general graphs, which will lead to a large amount of information loss. Secondly, its data augmentation strategy does not consider the characteristics of hypergraphs, so it cannot obtain high-level information through general methods.

To solve these problems, HyperGCL (Wei et al., 2022) explores various view augmentation strategies, such as hyperedge perturbation (Li et al., 2022), node masking (Hassani & Khasahmadi, 2020), and feature masking (You et al., 2020). However, due to random masking strategies, its results show no significant advantages of feature masking over node masking or hyperedge perturbation. TriCL (Lee & Shin, 2022) proposes a "three-way" contrastive learning method, considering node, group, and member level comparisons. MMACL (Lee & Chae, 2024) explores a mixed-attention method combining hypergraph and graph structures. LFH (Zhang et al., 2025) and DH (Hayat et al., 2024) note heterogeneity's impact, but these methods propose solutions only based on labeled data, lacking unsupervised solutions. However, traditional hypergraph contrastive learning often relies on random masking to generate contrastive views, which may not fully exploit domain structural properties.

## 3. Preliminaries

Given a hypergraph $G = (V, E)$, the incidence matrix $H \in \{0, 1\}^{|V| \times |E|}$ is defined as:

$$h(v, e) = \begin{cases} 1, & \text{if } v \in e, \\ 0, & \text{if } v \notin e. \end{cases}$$

The incidence matrix captures the structural relationships in the hypergraph by indicating which nodes are contained in which hyperedges.

We use the diagonal matrix $D_V$ to represent the degree of vertices, where its entries $d_v^i = \sum_j h_{ij}$. We also use the diagonal matrix $D_E$ to denote the degree of hyperedges, where its element $d_e^j = \sum_i h_{ij}$ represents the number of nodes connected by the hyperedge $e_j$.

To quantify the heterogeneity degree of a specific hyper-

graph dataset, we propose a hyperedge-guided hypergraph heterogeneity estimation method. Hyperedges serve as bridges for node information transmission and can be used to calculate hypergraph heterogeneity. We use the cosine similarity between the centroid of node features within a hyperedge and individual node features to measure the heterogeneity degree of each hyperedge. The average value across all hyperedges is then taken as the degree of heterogeneity of the hypergraph. This method captures the heterogeneity between nodes and the hyperedge centroid. The specific calculation formula is as follows:

$$x_e = \frac{1}{|e|} \sum_{v \in e} x_v, \tag{1}$$

$$\text{HD} = \frac{1}{|E|} \sum_{e \in E} \left( \frac{1}{|e|} \sum_{v \in e} \left( 1 - \frac{x_v \cdot x_e}{\|x_v\| \|x_e\|} \right) \right). \tag{2}$$

Based on the homogeneity assumption, most hypergraph contrastive learning methods assume that the neighborhoods of nodes are homogeneous, whether in the view augmentation stage or the feature aggregation stage. According to the method we proposed for calculating the heterogeneity degree of hypergraphs, see Equation (2), we have calculated the heterogeneity degrees of common hypergraph datasets. Specifically, the calculated heterogeneity scores are: 20news (0.4342), Cora-CoCitation (0.3354), Cora-CoAuthor (0.3329), Pubmed (0.2990), Citeseer (0.2984), and Zoo (0.1481). In contrast, the point cloud dataset ModelNet40 (Wu et al., 2015) (0.0482), the NTU dataset (Chen et al., 2003) (0.0371), and the classification dataset Mushroom (Dua & Graff, 2017) (0.0570) exhibit relatively low heterogeneity. This quantitative analysis confirms that, with the exception of specific domain datasets, most benchmarks used in previous studies exhibit high degrees of heterogeneity.

Moreover, even in datasets with a relatively low degree of heterogeneity, there are still heterogeneous points that raise the level of heterogeneity. In hypergraph neural networks, aggregating information from heterogeneous nodes can degrade node feature representation. Therefore, we aim to propose a heterogeneity-aware hypergraph contrastive learning model to reduce the influence of heterogeneous nodes on node feature representation.

## 4. Method

We aim to obtain a hypergraph encoder with heterogeneous relationship awareness. It captures heterogeneous information via hyperedges, identifies heterogeneous relationships from input node features, processes them during view augmentation and feature aggregation stages, mitigates their influence on node representations, thus obtaining better feature representations. The overview is shown in Figure 2.

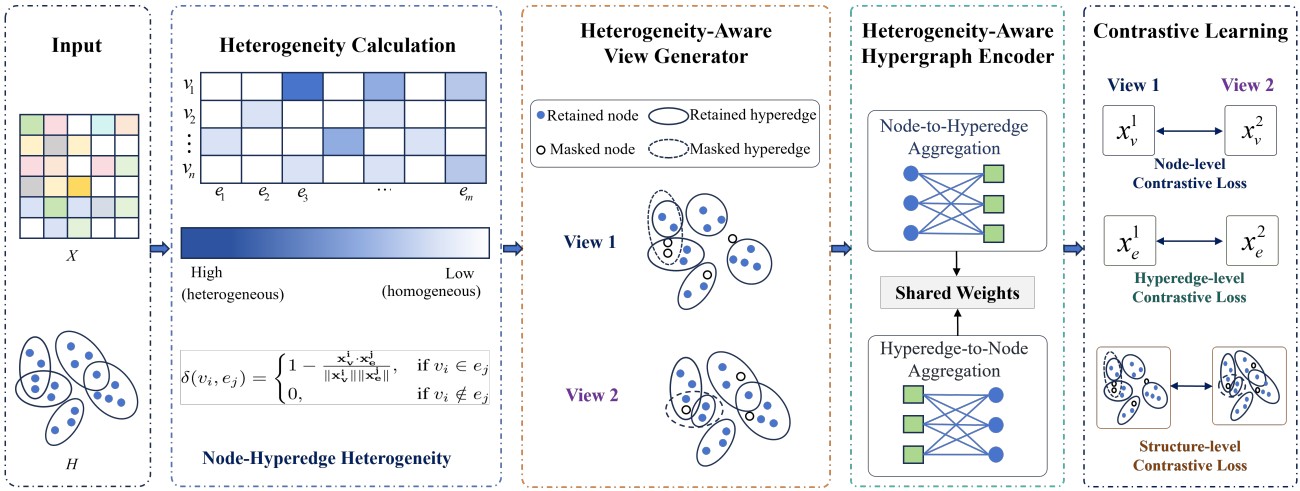

*Figure 2.* Overview of our proposed method. First, node–hyperedge heterogeneity is computed using hyperedges as structural units. In the Heterogeneity-Aware View Generator, high-heterogeneity incidences are more likely to be masked, while low-heterogeneity incidences are retained with higher probability. The heterogeneity matrix further guides feature selection and filtering in the Heterogeneity-Aware Hypergraph Encoder. Finally, a three-level contrastive objective is used for training.

### 4.1. Heterogeneity Calculation

We leverage hyperedges as bridges for node information propagation to capture heterogeneous information. Using input features, we compute node-hyperedge heterogeneity to evaluate neighborhood heterogeneity, with each pair having a heterogeneity value. Since hyperedges lack intrinsic features, they represent high-order structures. We use node features' average per hyperedge as its feature. Specifically, we compute hyperedge degrees $D_e$ and use hypergraph incidence matrix $H$ to calculate normalized matrix $N_{n2e}$ for initial hyperedge feature representation. The formulas are:

$$N_{n2e} = HD_e^{-1}, \quad (3)$$

$$\mathbf{x}_{e_j} = \sum_{v_i \in e_j} N_{n2e}^{ij} \mathbf{x}_{v_i}. \quad (4)$$

Considering that node features depend more on direction than magnitude, we compute node-hyperedge heterogeneity using cosine distance (focusing on feature vector angles). Compared to Jaccard, Manhattan, and Euclidean distances in downstream tasks, cosine distance achieved superior performance. See Appendix B for details. The heterogeneity $\delta(v_i, e_j)$ between node $v_i$ and hyperedge $e_j$ is calculated as follows:

$$\delta(v_i, e_j) = \begin{cases} 1 - \frac{\mathbf{x_v^i} \cdot \mathbf{x_e^j}}{\|\mathbf{x_v^i}\|\|\mathbf{x_e^j}\|}, & \text{if } v_i \in e_j, \\ 0, & \text{if } v_i \notin e_j, \end{cases} \quad (5)$$

where $\mathbf{x_v^i}$ denotes the feature of node $v_i$ belonging to hyperedge $e_j$, and $\mathbf{x_e^j}$ represents the feature of hyperedge $e_j$.

### 4.2. Heterogeneity-Aware View Generator

Heterogeneous nodes in hypergraphs induce feature representation bias. Contrastive learning mitigates this by enhancing cross-view consistency and suppressing noisy/heterogeneous information. However, traditional hypergraph contrastive methods rely on random masking, which only passively reduces interference. To effectively suppress the influence of heterogeneous nodes, instead of random node masking, we introduce heterogeneity-aware masking coefficients to ensure that highly heterogeneous nodes are more likely to be masked and their contributions to different views are further weakened. This enhances consistency between augmented views and facilitates learning of more homogeneous information from neighborhoods.

*Heterogeneity-Aware Incidence Matrix Mask*: After calculating the heterogeneity $\delta(v_i, e_j)$ between nodes and hyperedges, we normalize it into a retention score $r_{v_i, e_j}$ and use the complementary value as the masking probability $m_{v_i, e_j}$:

$$r_{v_i, e_j} = \frac{\delta_{\max} - \delta(v_i, e_j)}{\delta_{\max} - \delta_{\min} + \epsilon}, \quad m_{v_i, e_j} = 1 - r_{v_i, e_j}, \quad (6)$$

where $\delta_{\max}$ and $\delta_{\min}$ are the maximum and minimum heterogeneity values over all incident node-hyperedge pairs, and $\epsilon$ prevents division by zero. Hence, highly heterogeneous incidences obtain lower retention scores and are more likely to be masked.

To prevent deterministic augmentation, we implement multi-round stochastic sampling. Each iteration samples a temporary drop indicator per pair using the current masking probabilities, progressively approaching the target masking ratio:

$$d_{v_i, e_j}^{\text{temporary}} = \text{Bernoulli}(m_{v_i, e_j}). \quad (7)$$

The temporary masking matrix is dynamically adjusted using cumulative prior masking results, progressively converging to the target ratio. Once the predefined ratio is achieved, the resulting retention matrix is applied to the hypergraph structure for view construction. This process balances heterogeneity-aware filtering and stochastic diversity.

### 4.3. Heterogeneity-Aware Hypergraph Encoder

We design a shared-parameter encoder that uses hyperedges as filtering hubs. Node features aggregate via hyperedges, adaptively amplifying homogeneous aggregation while suppressing heterogeneous features. This focuses hyperedges on homogeneous information. The processed features are fed back to nodes, which adaptively aggregate hyperedge information by heterogeneity levels. This dual-stage homogeneity-weighted aggregation reduces heterogeneous interference, yielding purer homogeneous representations.

*Heterogeneity-Aware Normalized Matrix Update.* To utilize heterogeneous information $\delta(v_i, e_j)$ captured by hyperedges in feature aggregation, we perform heterogeneity-aware updates on initial normalization matrices $N_{n2e}$ and $N_{e2n}$ at each aggregation layer. For each hyperedge $e_j$ containing node $v_i$, we update and apply two-stage coefficients $\widetilde{N}_{n2e}^{(ij)}$ and $\widetilde{N}_{e2n}^{(ij)}$ for node-to-hyperedge and hyperedge-to-node, respectively:

$$\widetilde{N}_{n2e}^{(ij)} = \frac{\exp(1 - \delta(v_i, e_j))}{\sum_{v_k \in e_j} \exp(1 - \delta(v_k, e_j))}, \quad (8)$$

$$\widetilde{N}_{e2n}^{(ij)} = \frac{\exp(1 - \delta(v_i, e_j))}{\sum_{e_k \ni v_i} \exp(1 - \delta(v_i, e_k))}. \quad (9)$$

Based on $\widetilde{N}_{n2e}^{(ij)}$ (8) and $\widetilde{N}_{e2n}^{(ij)}$(9), we obtain normalization matrices $\widetilde{N}_{n2e}$ and $\widetilde{N}_{e2n}$ containing heterogeneous information. For each node-hyperedge pair, we use the natural exponential function to amplify heterogeneity's influence. Lower heterogeneity yields smaller $\delta(v_i, e_j)$ but larger $\widetilde{N}_{n2e}^{(ij)}$ and $\widetilde{N}_{e2n}^{(ij)}$, giving greater aggregation weights; higher heterogeneity reduces weights. This enables hyperedges to serve as intermediate structures for updating node features, capturing and filtering heterogeneous information.

*Heterogeneity-Aware Feature Aggregation.* After obtaining the normalization matrices $\widetilde{N}_{n2e}$ and $\widetilde{N}_{e2n}$ containing heterogeneous information, the model updates the node feature representations through two-stage feature aggregation. In this process, hyperedges serve as intermediate structures for information transfer, dynamically and selectively aggregating features from various nodes and suppressing information with higher heterogeneity to achieve efficient capture of homogeneous information.

The first stage is the aggregation of information from nodes to hyperedges. In this stage, the model aggregates the node features in each hyperedge according to the heterogeneity-aware normalization coefficients $\widetilde{N}_{n2e}$. The $(l+1)$-th layer hyperedge features $x_{e_j}^{(l+1)}$ are represented as:

$$x_{e_j}^{(l+1)} = \sum_{v_i \in e_j} \widetilde{N}_{n2e}^{(ij)} x_{v_i}^{(l)} W_{n2e}^{(ij)}, \quad (10)$$

where $x_{v_i}^{(l)}$ represents the features of node $v_i$ in the $l$-th layer, and $W_{n2e}^{(ij)}$ is the model-learnable parameter for the change from node to hyperedge features.

Next, nodes adaptively weight and update according to the heterogeneity-aware normalization coefficients $\widetilde{N}_{e2n}$ for each hyperedge feature. The features of the node of the $(l+1)$-th layer $x_{v_i}^{(l+1)}$ are represented as:

$$x_{v_i}^{(l+1)} = \sum_{e_j \ni v_i} \widetilde{N}_{e2n}^{(ij)} x_{e_j}^{(l+1)} W_{e2n}^{(ij)}. \quad (11)$$

Then we obtain the node feature update formula with heterogeneous awareness. The overall node feature update formula is as follows:

$$X^{(l+1)} =$$

$$\sigma \left( \sum_{v \in V} \widetilde{N}_{e2n} \left( \sum_{e \in E} \widetilde{N}_{n2e} X^{(l)} W_{n2e} + b_{n2e} \right) W_{e2n} + b_{e2n} \right),$$

where $X^{(l+1)}$ denotes the node features of the $(l+1)$-th layer, $W_{n2e}$ and $W_{e2n}$ are learnable parameters from nodes to hyperedges and from hyperedges to nodes, respectively. $\sigma$ is activation function, and $b_{n2e}$ and $b_{e2n}$ are bias terms.

### 4.4. Contrastive Learning

After heterogeneity-aware view generation, two views of the hypergraph structure, $G_1$ and $G_2$, are generated. The relationships between hyperedges and nodes in the input hypergraph $G$ undergo certain variations, and the features of nodes and hyperedges are updated accordingly. Based on these variations, we devise a dedicated contrastive loss function tailored to hypergraph representation learning.

*Node-level Contrastive Loss.* The node-level contrastive loss is defined as:

$$\ell_v(x_v^{1,i}, x_v^{2,i}) = -\log \frac{\exp(s(x_v^{1,i}, x_v^{2,i})/\tau_v)}{\sum_{k=1}^{|V|} \exp(s(x_v^{1,i}, x_v^{2,k})/\tau_v)},$$

where $x_v^{1,i}$ and $x_v^{2,i}$ denote the features of node $i$ in views $G_1$ and $G_2$, respectively. The function $s(x_v^{1,i}, x_v^{2,k})$ represents the cosine similarity between node $i$ in view $G_1$ and node $k$ in view $G_2$, while $\tau_v$ is the temperature coefficient. The node-level contrastive loss aims to align representations of

the same node across two views while separating different nodes' representations, increasing feature discrimination. We average the node-level loss across all positive samples:

$$L_v = \frac{1}{2|V|} \sum_{i=1}^{|V|} \left\{ \ell_v(x_v^{1,i}, x_v^{2,i}) + \ell_v(x_v^{2,i}, x_v^{1,i}) \right\}.$$

*Hyperedge-Level Contrastive Loss.* The hyperedge-level contrastive loss is defined as follows:

$$\ell_e(x_e^{1,j}, x_e^{2,j}) = -\log \frac{\exp(s(x_e^{1,j}, x_e^{2,j})/\tau_e)}{\sum_{k=1}^{|E|} \exp(s(x_e^{1,j}, x_e^{2,k})/\tau_e)},$$

where $x_e^{1,j}$ and $x_e^{2,j}$ represent the features of hyperedge $j$ in views $G_1$ and $G_2$, respectively. The overall hyperedge-level contrastive loss is then given by:

$$L_E = \frac{1}{2|E|} \sum_{j=1}^{|E|} \left\{ \ell_e(x_e^{1,j}, x_e^{2,j}) + \ell_e(x_e^{2,j}, x_e^{1,j}) \right\}.$$

*Structural-Level Contrastive Loss.* We define a bilinear scorer $A(x_v^i, x_e^j)$ to evaluate the compatibility score that a node $v_i$ belongs to a hyperedge $e_j$, formulated as:

$$A(x_v^i, x_e^j) = (x_v^i)^T W x_e^j + b,$$

where $W$ is a learnable parameter matrix and $b$ is a bias term. The loss function when the node $v_i$ is the anchor is defined as:

$$Z_i^j = \exp(A(x_v^i, x_e^j)/\tau_s)$$
$$+ \sum_{k \neq j} \exp(A(x_v^i, x_e^k)/\tau_s),$$
$$\ell_s(x_v^i, x_e^j) = -\log \frac{\exp(A(x_v^i, x_e^j)/\tau_s)}{Z_i^j}.$$

The overall structural-level loss is computed as the average loss across all observed node–hyperedge incidences, as follows:

$$L_S = \frac{1}{|\Omega|} \sum_{(v_i, e_j) \in \Omega} \ell_s(x_v^i, x_e^j),$$

where $\Omega$ denotes the set of valid node–hyperedge connections. In implementation, non-incident hyperedges in $Z_i^j$ are sampled as negatives for efficiency.

*Overall Loss Function.* The final overall loss combines the three levels of contrastive losses (node-level, hyperedge-level, and structural-level) with weights, as:

$$L = w_v L_v + w_E L_E + w_S L_S,$$

where $w_v$, $w_E$, and $w_S$ are weighting coefficients that balance the contribution of each loss term.

## 4.5. Theoretical Analysis

In this section, we provide a theoretical analysis of the proposed H²CL method. We analyze the model from two perspectives: (1) we prove that our encoder minimizes a heterogeneity-weighted Dirichlet energy, effectively enforcing smoothness among homogeneous nodes while allowing cuts at heterogeneous boundaries; and (2) we show that our heterogeneity-aware view generation reduces false-positive retention under a heterogeneity-semantic calibration assumption.

Standard hypergraph propagation often leads to over-smoothing by treating all connections equally. We prove that the propagation mechanism in our Heterogeneity-Aware Hypergraph Encoder is equivalent to minimizing a specific energy function that penalizes differences between nodes and hyperedges, strictly weighted by their homogeneity.

**Definition 4.1** (Heterogeneity-Weighted Dirichlet Energy). Consider the hypergraph $G = (V, E)$ with node features $Z_v$ and hyperedge features $Z_e$. Let $\mathcal{W}_{v,e} = \exp(1 - \delta(v, e))$ be the homogeneity weight derived from the heterogeneity measure $\delta$. We define the Heterogeneity-Weighted Dirichlet Energy $E_{HG}$ as:

$$E_{HG}(Z_v, Z_e) = \sum_{v \in V} \sum_{e \in E} \mathcal{W}_{v,e} \|z_v - z_e\|^2.$$

Intuitively, minimizing $E_{HG}$ forces nodes to be close to the hyperedges they belong to. However, unlike standard propagation, the weight $\mathcal{W}_{v,e}$ acts as a gate: high heterogeneity yields low weights, allowing the model to cut the connection in feature space and preventing the propagation of noise.

**Theorem 4.2.** *The update rule of the Heterogeneity-Aware Hypergraph Encoder constitutes a coordinate descent step that minimizes $E_{HG}$.*

*Proof.* The objective function can be rewritten in matrix form as:

$$E_{HG} = \text{tr}(Z_v^T D_v Z_v) + \text{tr}(Z_e^T D_e Z_e) - 2\text{tr}(Z_v^T \mathbf{W} Z_e),$$

where $\mathbf{W} \in \mathbb{R}^{|V| \times |E|}$ is the weighted incidence matrix with entries $\mathcal{W}_{v,e}$, and $D_v, D_e$ are diagonal degree matrices where $(D_v)_{ii} = \sum_e \mathcal{W}_{v_i,e}$ and $(D_e)_{jj} = \sum_v \mathcal{W}_{v,e_j}$. Taking the partial derivative with respect to $Z_e$ and setting it to 0 to find the optimal hyperedge representation given current nodes:

$$\frac{\partial E_{HG}}{\partial Z_e} = 2D_e Z_e - 2\mathbf{W}^T Z_v = 0 \implies Z_e = D_e^{-1} \mathbf{W}^T Z_v.$$

This corresponds exactly to our node-to-hyperedge aggregation step (Eq. 8), where $\widetilde{N}_{n2e} \approx D_e^{-1} \mathbf{W}^T$. Substituting the optimal $Z_e$ back into the derivative with respect to $Z_v$:

$$\frac{\partial E_{HG}}{\partial Z_v} = 2D_v Z_v - 2\mathbf{W} Z_e = 0 \implies Z_v = D_v^{-1} \mathbf{W} Z_e.$$

This matches our hyperedge-to-node aggregation (Eq. 9). Thus, our encoder iteratively minimizes the weighted energy, clustering semantically similar nodes while ignoring heterogeneous outliers. $\square$

Noise Suppression. From a spectral graph theory perspective, the transition matrix defined by our method is $\mathbf{P} = D_v^{-1} \mathbf{W} D_e^{-1} \mathbf{W}^T$. In standard hypergraph learning, the incidence matrix $H$ often connects nodes of different classes (heterogeneity), creating "high-frequency" paths that propagate noise. In H$^2$CL, by assigning low weights $\mathcal{W}_{v,e}$ to heterogeneous pairs, the transition probability between semantically different classes approaches zero. Consequently, the propagation operator $\mathbf{P}$ approximates a block-diagonal matrix corresponding to semantic clusters. This acts as a semantic low-pass filter, suppressing high-frequency interclass noise while preserving low-frequency intra-class information, thereby improving the signal-to-noise ratio of the representation.

False Positive Reduction. The core objective of our framework utilizes the InfoNCE loss, which optimizes for *Alignment* (closeness of positive pairs) and *Uniformity* (separation of negative pairs). A critical failure mode in hypergraph contrastive learning is the "False Positive" problem: random masking may generate a view $v'$ that is structurally connected to $v$ via a hyperedge but belongs to a different semantic class (due to heterogeneity). Forcing alignment on such pairs $(sim(v, v') \uparrow)$ corrupts the embedding space. Our Heterogeneity-Aware View Generator uses retention scores $r_{v,e}$ and masking probabilities $m_{v,e} = 1 - r_{v,e}$ to guide augmentation. By masking highly heterogeneous incidences with high probability, the retained structure in views $G_1$ and $G_2$ is biased toward homogeneous connections. Appendix D.2 formalizes this claim: under a heterogeneity-semantic calibration assumption, the induced retention distribution has lower KL divergence to the clean semantic incidence distribution and lower false-positive retention risk than uniform masking.

# 5. Experiments

In this section, we conduct a comprehensive evaluation of H$^2$CL to verify its effectiveness in learning robust node representations. Our evaluation aims to address three core research questions: first, we assess whether H$^2$CL outperforms state-of-the-art graph and hypergraph representation learning baselines, particularly in node classification tasks (RQ1); second, we investigate whether H$^2$CL can maintain stable performance when facing structural noise and varying degrees of heterogeneity compared to existing methods (RQ2); and third, we analyze how the proposed Heterogeneity-Aware View Generator and Hypergraph Encoder individually contribute to the model's success through a detailed ablation study (RQ3).

## 5.1. Experimental Setup

**Datasets and Baselines.** We utilize a diverse set of real-world benchmarks covering citation, co-citation, and co-authorship networks (Cora-CC, Cora-CA, Citeseer, PubMed), computer vision datasets (NTU, ModelNet40), and UCI/text datasets (20News, Mushroom). Zoo is used for robustness and heterogeneity-measurement analysis, while Actor, Walmart, and Trivago provide additional larger-scale evaluation in Section 5.3. These datasets exhibit varying scales and degrees of heterogeneity, providing a rigorous testbed for evaluating model generalizability across different domains. To ensure a rigorous comparison, we evaluate H$^2$CL against 13 representative baselines classified into three distinct categories. The first category includes Graph Neural Networks (GNNs) such as GCN (Kipf & Welling, 2017) and GAT (Veličković et al., 2018a), which treat hypergraphs as clique-expanded simple graphs. The second category comprises supervised Hypergraph Neural Networks, including HGNN (Feng et al., 2019), HyperGCN (Yadati et al., 2019), HNHN (Dong et al., 2020), HyperConv (Bai et al., 2021), UniGCN (Huang & Yang, 2021), and AllSet (Chien et al., 2022), which rely on label supervision and standard message passing. The third category focuses on Self-Supervised and Contrastive Learning methods, ranging from general graph approaches like DGI (Veličković et al., 2018b) and GRACE to state-of-the-art hypergraph contrastive frameworks such as $S^2$-HHGR, TriCL (Lee & Shin, 2022) and CHGNN (Song et al., 2024). **Implementation details are described in Appendix C.**

## 5.2. Performance Analysis

Table 1 summarizes the node classification performance, revealing critical insights regarding the superiority of our approach (RQ1). First, H$^2$CL consistently outperforms topology-based GNNs (GCN, GAT) across all datasets. For instance, on ModelNet40, H$^2$CL achieves 97.40% accuracy, significantly surpassing GCN's 91.67%. This margin validates the necessity of preserving high-order structures rather than degrading them into pairwise cliques, which leads to information loss.

Furthermore, compared to supervised HGNNs, our self-supervised framework achieves competitive or superior results. This is notable given that H$^2$CL does not utilize labels during feature learning. The performance gap is most evident on high-heterogeneity datasets such as 20News, where spectral methods like HyperGCN struggle due to their inability to account for feature contamination—a limitation effectively addressed by our heterogeneity-aware filtering.

Finally, H$^2$CL surpasses the leading baseline, TriCL, on all seven main datasets. While TriCL utilizes group-level

*Table 1.* Node classification accuracy and standard deviations. For each dataset, the best and the second-best performances are highlighted in **boldface** and underlined, respectively. OOM indicates out of memory on a 24GB GPU.

| Method | Cora-CC | Cora-CA | Citeseer | 20News | Mushroom | NTU | ModelNet40 |
|---|---|---|---|---|---|---|---|
| GCN | $77.11 \pm 1.8$ | $73.66 \pm 1.3$ | $66.07 \pm 2.4$ | OOM | $92.47 \pm 0.9$ | $71.17 \pm 2.4$ | $91.67 \pm 0.2$ |
| GAT | $77.75 \pm 2.1$ | $74.52 \pm 1.0$ | $67.62 \pm 3.5$ | OOM | OOM | $70.94 \pm 2.6$ | $91.43 \pm 0.3$ |
| HGNN | $77.50 \pm 1.8$ | $74.38 \pm 1.2$ | $66.16 \pm 2.3$ | $\underline{80.15 \pm 0.3}$ | $89.58 \pm 0.5$ | $72.03 \pm 2.4$ | $92.23 \pm 0.2$ |
| HyperConv | $76.19 \pm 2.1$ | $73.52 \pm 0.7$ | $64.12 \pm 2.6$ | $79.83 \pm 0.4$ | $97.56 \pm 0.6$ | $72.62 \pm 2.6$ | $91.84 \pm 0.1$ |
| HNHN | $76.21 \pm 1.7$ | $74.88 \pm 1.6$ | $67.28 \pm 2.2$ | $79.51 \pm 0.4$ | $99.75 \pm 0.1$ | $71.45 \pm 3.2$ | $92.96 \pm 0.2$ |
| HyperGCN | $64.11 \pm 7.4$ | $60.65 \pm 8.3$ | $59.92 \pm 9.6$ | $77.31 \pm 6.0$ | $48.26 \pm 0.3$ | $46.05 \pm 3.9$ | $69.23 \pm 2.8$ |
| UniGCN | $77.91 \pm 1.9$ | $77.30 \pm 1.4$ | $66.40 \pm 1.9$ | $\mathbf{80.24 \pm 0.4}$ | $98.84 \pm 0.5$ | $73.27 \pm 2.7$ | $94.62 \pm 0.2$ |
| AllSet | $76.21 \pm 1.7$ | $76.94 \pm 1.3$ | $67.83 \pm 1.8$ | $79.90 \pm 0.4$ | $\underline{99.78 \pm 0.1}$ | $\underline{75.09 \pm 2.5}$ | $96.85 \pm 0.2$ |
| DGI | $78.17 \pm 1.4$ | $76.94 \pm 1.1$ | $68.81 \pm 1.8$ | OOM | OOM | $72.01 \pm 2.5$ | $92.18 \pm 0.2$ |
| GRACE | $79.11 \pm 1.7$ | $76.59 \pm 1.0$ | $68.65 \pm 1.7$ | OOM | OOM | $70.51 \pm 2.4$ | $90.68 \pm 0.3$ |
| S$^2$-HHGR | $78.08 \pm 1.7$ | $78.15 \pm 1.1$ | $68.21 \pm 1.8$ | $79.75 \pm 0.3$ | $97.15 \pm 0.5$ | $73.95 \pm 2.4$ | $93.26 \pm 0.2$ |
| TriCL | $\underline{81.37 \pm 1.1}$ | $\underline{82.00 \pm 0.6}$ | $\underline{71.82 \pm 1.2}$ | $79.84 \pm 0.2$ | $99.63 \pm 0.1$ | $74.96 \pm 2.4$ | $97.08 \pm 0.1$ |
| CHGNN | $79.8 \pm 1.0$ | $78.9 \pm 0.7$ | $69.5 \pm 1.1$ | $79.87 \pm 0.2$ | $99.70 \pm 0.1$ | $73.90 \pm 0.2$ | $\underline{97.10 \pm 0.1}$ |
| H$^2$CL (Ours) | $\mathbf{81.67 \pm 1.2}$ | $\mathbf{82.02 \pm 0.8}$ | $\mathbf{72.17 \pm 1.4}$ | $80.05 \pm 0.2$ | $\mathbf{99.84 \pm 0.1}$ | $\mathbf{75.27 \pm 2.7}$ | $\mathbf{97.40 \pm 0.1}$ |

contrast, it treats all members within a group equally, ignoring the semantic noise introduced by heterogeneous nodes. In contrast, H$^2$CL's ability to distinctively weight interactions based on node-hyperedge heterogeneity allows it to learn more discriminative embeddings, thereby validating the effectiveness of our proposal in complex scenarios.

### 5.3. Additional Evaluation and Scalability

Table 2 complements the main benchmark results with PubMed, recent hypergraph baselines, and larger public hypergraphs. On PubMed, H$^2$CL achieves 87.92%, outperforming TriCL, CHGNN, Kernelized HGNNs, and Adaptive Expansion. To examine scalability, we further evaluate on Actor, Walmart, and Trivago from DHG-Bench (Li et al., 2026), which contain 16,255/10,164, 88,860/69,906, and 172,738/233,202 nodes/hyperedges, respectively. H$^2$CL achieves the best accuracy on all three datasets, showing that the heterogeneity-aware design remains effective beyond the smaller benchmarks in Table 1. The following subsection reports loss-ablation and masking-ratio analyses; Section 5.6 further inspects learned reweighting behavior, and Appendix D gives complexity and view-generation analyses.

### 5.4. Loss Components and Masking-Ratio Analyses

Table 3 studies two important design choices. The full three-level objective consistently outperforms single-loss and pairwise-loss variants, with the node-level loss providing the strongest single signal and the structural loss acting as a useful regularizer when combined with the other terms. H$^2$CL is also stable across masking ratios from 0.2 to 0.5, while overly small ratios leave heterogeneous incidences insufficiently suppressed and overly large ratios remove useful

homogeneous information.

### 5.5. Robustness and Ablation Analysis

To address RQ2 and RQ3, we conducted controlled experiments by injecting structural noise, adding non-member nodes to hyperedges at varying ratios (from 10% to 70%) to simulate increasing heterogeneity. We compared four model variations: standard HGNN (red curve), HGNN + Generator (green curve), HGNN + Encoder (orange curve), and the full H$^2$CL model (blue curve).

The results on Cora, Citeseer, and Zoo, plotted in Figure 4 (See Appendix A), illustrate a clear trend. As the noise ratio increases, the performance of the standard HGNN drops precipitously, confirming our hypothesis that indiscriminate aggregation makes standard models highly vulnerable to feature contamination. In contrast, both HGNN + Generator and HGNN + Encoder exhibit significantly improved stability. The View Generator effectively prevents the model from learning "false positive" structural correlations during pre-training, while the Encoder acts as a runtime filter to block noisy message passing. Most importantly, the full H$^2$CL model consistently achieves the highest accuracy and the lowest degradation rate across all noise levels. This demonstrates a synergistic benefit: filtering heterogeneity at both the data augmentation stage and the propagation stage provides mutually reinforcing protection, enabling the model to extract high-quality signals even from highly contaminated structures.

To determine the most effective metric for quantifying the heterogeneity $\delta(v, e)$ between nodes and hyperedges, we evaluated four distinct distance measures: Cosine distance, Dot product, Manhattan distance ($L_1$), and Euclidean distance ($L_2$). **See Appendix B.**

*Table 2.* Additional node-classification results on recent baselines, and larger hypergraphs (Accuracy %).

*(a)* Recent baselines.

| Method | Cora-CA | Citeseer | PubMed |
|---|---|---|---|
| TriCL (Lee & Shin, 2022) | 82.00 | 71.82 | 84.26 |
| CHGNN (Song et al., 2024) | 78.90 | 69.50 | 79.00 |
| Kernelized HGNNs (Feng et al., 2025) | 79.69 | 69.61 | 77.41 |
| Adaptive Expansion (Ma et al., 2025) | 81.42 | 70.46 | 86.82 |
| H$^2$CL (Ours) | **82.02** | **72.17** | **87.92** |

*(b)* Larger hypergraphs.

| Method | Actor | Walmart | Trivago |
|---|---|---|---|
| HGNN | 77.83 | 77.12 | 57.67 |
| HNHN | 81.20 | 65.21 | 53.75 |
| H$^2$CL (Ours) | **86.22** | **78.57** | **85.72** |

*Table 3.* Ablation and sensitivity analyses on the three contrastive losses and the masking ratio (Accuracy %).

*(a)* Contrastive loss components.

| Loss | Cora-CC | Citeseer | 20News |
|---|---|---|---|
| $L_v$ only | 79.24 | 69.87 | 78.35 |
| $L_E$ only | 78.91 | 69.15 | 77.92 |
| $L_S$ only | 77.53 | 68.02 | 76.44 |
| $L_v + L_E$ | 80.46 | 70.93 | 79.11 |
| $L_v + L_S$ | 79.87 | 70.21 | 78.69 |
| $L_E + L_S$ | 79.02 | 69.58 | 78.02 |
| $L_v + L_E + L_S$ | **81.67** | **72.17** | **80.05** |

*(b)* Masking-ratio sensitivity.

| Ratio | Cora-CC | Citeseer | 20News |
|---|---|---|---|
| 0.1 | 80.52 | 70.84 | 78.96 |
| 0.2 | 81.03 | 71.46 | 79.58 |
| 0.3 | **81.67** | **72.17** | 80.05 |
| 0.4 | 81.42 | 71.89 | **80.12** |
| 0.5 | 80.85 | 71.23 | 79.74 |
| 0.6 | 80.17 | 70.52 | 78.91 |
| 0.7 | 79.23 | 69.64 | 78.03 |

## 5.6. Learned Reweighting Analysis

We further inspect whether the encoder learns to down-weight heterogeneous structures rather than merely adding another smoothing layer. On mixed Cora-CC hyperedges, H$^2$CL assigns an average aggregation weight of 0.45 to majority-class nodes but only 0.02 to minority semantic outliers, indicating that heterogeneous incidences are strongly suppressed during message passing. We also compute the effective heterogeneity degree over the learned transition matrix $P = D_v^{-1} W D_e^{-1} W^T$, where $W$ is the reweighted incidence matrix. As shown in Figure 3, the effective heterogeneity is substantially lower than that of the raw hypergraph on Cora-CC, Citeseer, and 20News, confirming that the learned information-flow topology is more homogeneous and better aligned with semantic clusters.

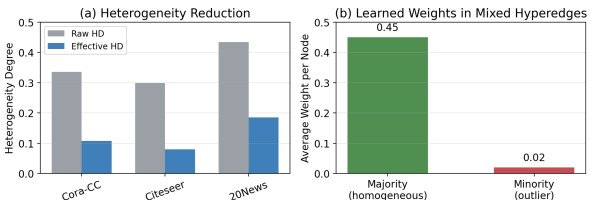

*Figure 3.* Learned reweighting behavior. Left: raw versus effective heterogeneity degree after H$^2$CL reweighting. Right: average aggregation weight per node in mixed Cora-CC hyperedges.

Together with Tables 2 and 3, this analysis links the empirical gains to the intended mechanisms: targeted augmentation reduces noisy positive pairs, three-level contrast supplies complementary supervision signals, and heterogeneity-aware propagation suppresses minority semantic outliers at the message-passing stage. The visualization also com-plements the robustness study by showing that the encoder changes the effective topology used for propagation, rather than only improving the classifier head. Thus the improvements are not only reflected in final accuracy, but also in a measurable reduction of heterogeneous information flow.

## 6. Conclusion

In this work, we introduced H$^2$CL, a heterogeneity-aware hypergraph contrastive learning framework, which suppresses noisy node–hyperedge interactions during both contrastive view construction and message passing. Theoretical analysis shows that the proposed encoder can be interpreted as minimizing a heterogeneity-weighted Dirichlet energy, providing a principled explanation for its ability to preserve homogeneous signals while weakening heterogeneous propagation. Extensive experiments on standard benchmarks and larger-scale hypergraphs demonstrate competitive performance compared with recent baselines. Further robustness, ablation, and learned reweighting analyses confirm that the two heterogeneity-aware components and the three-level contrastive objective jointly contribute to robust representation learning. These results suggest that explicit node–hyperedge heterogeneity modeling is a promising direction for reliable hypergraph representation learning.

## Acknowledgements

This work was supported by the National Natural Science Foundation of China (No. U25A20529, 62536006, 62406180), the Fundamental Research Program of Shanxi Province (No. 202403021212337).

## Impact Statement

This paper presents work whose goal is to advance the field of Machine Learning. There are many potential societal consequences of our work, none of which we feel must be specifically highlighted here.

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

# A. Robustness Curves Under Structural Noise

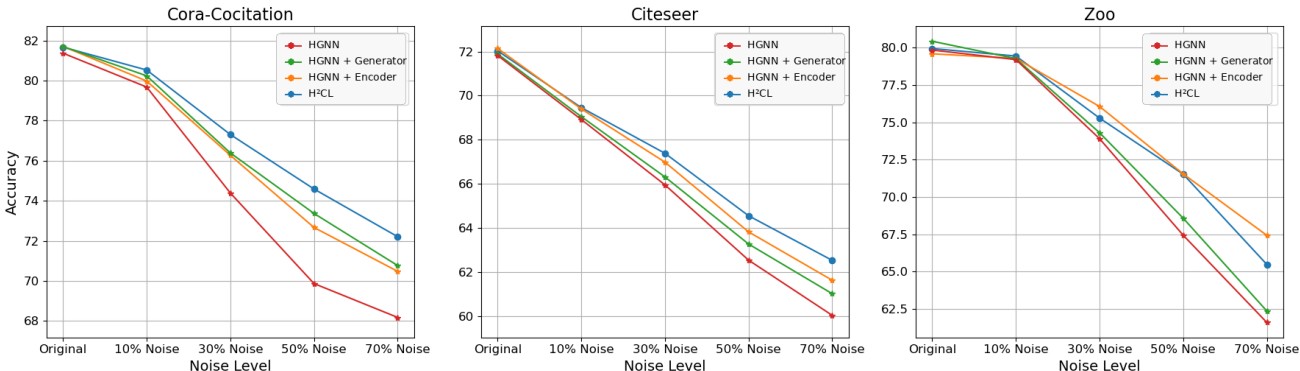

*Figure 4.* Performance degradation under different structural-noise ratios. The x-axis represents the proportion of added noisy nodes relative to the original hypergraph, while the y-axis indicates node classification accuracy. The four curves correspond to standard HGNN (red), HGNN + Generator (green), HGNN + Encoder (orange), and the full H²CL model (blue).

# B. Analysis of Different Heterogeneity Measurements

To determine the most effective metric for quantifying the heterogeneity $\delta(v, e)$ between nodes and hyperedges, we evaluated four distinct distance measures: Cosine distance, Dot product, Manhattan distance ($L_1$), and Euclidean distance ($L_2$). This ablation study was conducted across three datasets representing different modalities and heterogeneity levels: Cora-CoCitation (citation network), Zoo (UCI categorical data), and NTU (visual data).

**Theoretical Hypothesis.** We posit that in the high-dimensional latent spaces typically learned by Graph Neural Networks, the *semantic identity* of a node is primarily encoded in the *orientation* of its feature vector rather than its magnitude. Euclidean and Manhattan distances are sensitive to vector magnitude. In heterogeneous hypergraphs, nodes within the same semantic class may have varying feature magnitudes (e.g., due to different degrees or intensity), causing magnitude-based metrics to overestimate heterogeneity. In contrast, Cosine distance focuses solely on the angular alignment, which we hypothesize is a more robust proxy for semantic similarity.

**Quantitative Analysis.** The experimental results, summarized in Table 4, strongly support our hypothesis. Cosine distance consistently achieves the highest classification accuracy across all datasets (e.g., 81.67% on CoraCC). Notably, Euclidean distance suffers a catastrophic performance drop on the NTU dataset (42.31% vs. 75.27% for Cosine). This suggests that on visual datasets, feature magnitude acts as a confounding factor. Using Euclidean distance likely leads to "False Positive" heterogeneity detection—where semantically similar but magnitude-distinct nodes are incorrectly flagged as heterogeneous and filtered out—thereby stripping the model of valuable signal. Dot product, being an unnormalized version of Cosine similarity, also underperforms, further validating the necessity of normalization to ensure stability.

Based on these findings, we adopt Cosine distance as the standard metric for heterogeneity calculation in H²CL, as it provides the most robust guidance for our filtering mechanisms.

*Table 4.* Performance Comparison of Different Heterogeneity Measurement Methods (Accuracy %)

| Metric | Cora-CC | Zoo | NTU |
|---|---|---|---|
| Cosine (**Ours**) | **81.67** | **80.01** | **75.27** |
| Dot Product | 81.07 | 77.37 | 74.60 |
| Manhattan ($L_1$) | 81.33 | 75.63 | 74.44 |
| Euclidean ($L_2$) | 66.97 | 53.28 | 42.31 |

# C. Implementation Details

We adopt the standard linear evaluation protocol (Veličković et al., 2018b), first training the encoder unsupervised for node representations, then training a linear classifier on these frozen representations using L2-regularized logistic regression without gradient backpropagation. Datasets are randomly split into 10% training, 10% validation, and 80% test sets (Thakoor

et al., 2021). For stability, unsupervised experiments use 20 random splits with 10 weight initializations, while supervised experiments use 20 training splits with distinct parameters, reporting average classification accuracy.

## D. Additional Experiments and Analyses

This section provides supplementary analyses of complexity, view generation, and limitations. PubMed, recent-baseline, larger-scale, loss-ablation, masking-ratio, and learned-reweighting evaluations are included in the main text in Sections 5.3, 5.4, and 5.6.

### D.1. Complexity Analysis

Let $F$ be the input feature dimension, $d$ the embedding dimension, $|\mathcal{I}| = \text{nnz}(H)$ the number of nonzero incidences, $\bar{d}$ the average hyperedge degree, $B$ the mini-batch size, and $K$ the number of negative samples per anchor. Heterogeneity precomputation requires $O(|E|\bar{d}F)$ time, equivalently $O(|\mathcal{I}|F)$, and is performed once before training. Each heterogeneity-aware message-passing layer has complexity $O(|\mathcal{I}|d)$, matching the sparse incidence-matrix dependence of standard hypergraph neural networks up to the reweighting constants. The node-level and hyperedge-level contrastive losses require $O(BKd)$ time under negative sampling. A naive dense structural assessor would require $O(|V||E|d)$, but H$^2$CL computes positives only on valid incidences and samples $K$ disconnected hyperedges as negatives for each anchor, yielding $O(|\mathcal{I}_B|Kd)$ time and $O(BK)$ memory for a mini-batch incidence set $\mathcal{I}_B$.

### D.2. View Generator Alignment Analysis

Let $\Omega = \{(v,e) : v \in e\}$ and let $\delta(v,e)$ denote the node-hyperedge heterogeneity score. We explicitly distinguish the retention score and masking probability:

$$r_{v,e} = \frac{\delta_{\max} - \delta(v,e)}{\delta_{\max} - \delta_{\min} + \epsilon}, \qquad m_{v,e} = 1 - r_{v,e}. \tag{12}$$

Let $P^*(v,e)$ denote the latent clean semantic incidence distribution, i.e., semantically consistent incidences that should ideally be preserved. The heterogeneity-aware generator induces the retention distribution

$$Q_H(v,e) = \frac{r_{v,e}}{\sum_{(u,f)\in\Omega} r_{u,f}}, \tag{13}$$

while uniform random masking corresponds to $Q_U(v,e) = 1/|\Omega|$. Under the calibration assumption that semantically consistent incidences tend to have lower heterogeneity and therefore larger retention scores, $Q_H$ has lower cross-entropy with $P^*$ than $Q_U$. Therefore,

$$\begin{aligned} D_{\mathrm{KL}}(P^*\|Q_H) &= -H(P^*) + H(P^*, Q_H) \\ &\leq -H(P^*) + H(P^*, Q_U) = D_{\mathrm{KL}}(P^*\|Q_U). \end{aligned} \tag{14}$$

If $\Omega_{\text{het}} \subset \Omega$ denotes semantically inconsistent incidences, the false-positive retention risk $R_{\mathrm{FP}}(Q) = \sum_{(v,e)\in\Omega_{\text{het}}} Q(v,e)$ satisfies $R_{\mathrm{FP}}(Q_H) \leq R_{\mathrm{FP}}(Q_U)$ under the same calibration condition because $Q_H$ allocates less probability mass to high-heterogeneity incidences. This result does not claim universal recovery of the true semantic manifold; it shows that heterogeneity-aware retention improves semantic alignment over uniform masking when the heterogeneity score is semantically calibrated.

### D.3. Scope and Limitations

The assumption that heterogeneous incidences act as noise is scoped to node classification and semantic representation learning, where class separability is the goal. In tasks such as influence propagation or community bridging, heterogeneous connections may be essential signals rather than noise. In addition, H$^2$CL currently computes $\delta(v,e)$ once from the raw input features for efficiency and stability. This assumes that the raw features contain at least coarse semantic information. When features are heavily corrupted, missing, or purely topological, the heterogeneity scores may become nearly uniform; in that case the view generator naturally degrades to standard uniform masking rather than providing targeted filtering. A promising future direction is to update $\delta(v,e)$ dynamically from evolving embeddings during pre-training.

