# OpenReview forum: "H$^2$CL: Heterogeneity-Aware Hypergraph Contrastive Learning for Robust Representation Learning"
_ICML.cc/2026/Conference — ICML 2026 regular_

### Official Review · Reviewer_rLue · 2026-03-10

**Soundness:** 2
**Presentation:** 2
**Significance:** 2
**Originality:** 2
**Overall Recommendation:** 2
**Confidence:** 3

**Summary:**

In this paper, the authors propose a heterogeneity-aware hypergraph contrastive learning framework named HHCL. Specially, they argue that conventional hypergraph contrastive methods assume neighborhood homogeneity, which leads to feature contamination and false positive alignment in highly heterogeneous structures. To address this issue, they introduce a heterogeneity-aware view generator and a dynamic hypergraph encoder for selectively suppressing noisy node–hyperedge interactions during augmentation and propagation. Additionally, they also propose a three-level contrastive objective for enhancing the representation consistency. Extensive experiments on different benchmarks demonstrate the performance of the proposed HHCL .

**Compliance With Llm Reviewing Policy:**

Affirmed.

**Final Justification:**

After carefully checking the rebuttal again, I still have concerns for this paper: 1. The core assumption that heterogeneity should be treated as noise is not sufficiently justified. 2. The proposed method works as a reweighting-based smoothing scheme, with limited conceptual novelty. Moreover, the additional analyses in rebuttal mainly describe model behavior without providing clear evidence to answer my questions. Overall, the contribution and empirical evidence are not strong enough for acceptance, so I maintain my original score.

**Key Questions For Authors:**

Please see weaknesses

**Limitations:**

Please see weaknesses

**Strengths And Weaknesses:**

Strengths:

- This paper proposes a heterogeneity-aware hypergraph contrastive learning framework for explicitly modelling node–hyperedge heterogeneity, which improves the hypergraph representation learning in high-order and structurally complex settings.
- Theoretical analysis based on the Heterogeneity-Weighted Dirichlet Energy provides principled justification for the proposed encoder.
- Extensive experiments conducted on different benchmark datasets demonstrate the performance of the proposed HHCL.

Weaknesses:

- In introduction, the authors assume that hypergraph heterogeneity phenomenon act as sources of noise and leads to feature contamination. But, this assumption does not always hold in real-world applications. For example, in community detection hypergraph, bridge nodes may indeed introduce noise, but in influence propagation hypergraph, such bridge structures can be the key signal. I think the authors should more carefully justify this hypothesis.
- The core of HHCL is to perform heterogeneity-guided smoothing control via reweighting message passing, rather than explicitly modeling heterogeneous semantic structures, which is similar to [1]
- In the experimental part, authors claim they conducted experiments on Pubmed dataset, but no results are shown. Additionally, please consider more recent baselines [2,3,4].

[1] Ding K, Wang J, Li J, et al. Be more with less: Hypergraph attention networks for inductive text classification[C]//Proceedings of the 2020 conference on empirical methods in natural language processing (EMNLP). 2020: 4927-4936.

[2] Feng Y, Zhang Y, Ying S, et al. Kernelized hypergraph neural networks[J]. IEEE Transactions on Pattern Analysis and Machine Intelligence, 2025.

[3] Yi S, Mao Z, Wang Y, et al. Hypergraph consistency learning with relational distillation[J]. IEEE Transactions on Multimedia, 2025.

[4] Ma T, Qian Y, Zhang S, et al. Adaptive expansion for hypergraph learning[J]. arXiv preprint arXiv:2502.15564, 2025.

---

> ### Author Rebuttal · Authors · 2026-03-30
>
> We sincerely thank you for your thoughtful comments and for recognizing the strengths of our theoretical analysis and empirical results. We now address your concerns below:
>
> ■ **The scope of the Heterogeneity Assumption**
>
> We agree that the role of heterogeneous / bridge nodes is task-dependent. For special task, for example, influence maximization or community detection, bridge nodes are highly valuable signals rather than noise.
>
> Our assumption—that heterogeneous nodes cause feature contamination—is specifically grounded in the context of node classification and semantic representation learning, where the homophily assumption is the primary driver of class separability. In these scenarios, when a single hyperedge connects multiple distinct semantic communities, indiscriminate message passing dilutes the distinctiveness of the node embeddings.
>
> To empirically validate this within our problem scope, we quantified the actual heterogeneity of standard benchmark datasets using our proposed metric (Equation 2). As shown in the table below, the primary datasets used in our evaluation exhibit substantial heterogeneity:
>
> | Dataset | Heterogeneity Score |
> | :--- | :--- |
> | 20newsW100 | 0.4342 |
> | Cora-CoCitation | 0.3354 |
> | Cora-CoAuthor | 0.3329 |
> | Pubmed | 0.2990 |
> | Citeseer | 0.2984 |
> | Zoo | 0.1481 |
> | Mushroom | 0.0570 |
> | ModelNet40 | 0.0482 |
> | NTU2012 | 0.0371 |
>
> This analysis shows that several widely used classification benchmarks exhibit non-trivial node-hyperedge heterogeneity, which makes them relevant testbeds for our setting. We realize our scope was not clearly demarcated in the introduction. In the revised manuscript, we will explicitly clarify this boundary, including this table to demonstrate that $\mbox{H}^2\mbox{CL}$ is specifically effective to filter noise in high-heterogeneity, homophily-dependent tasks.
>
> ■ **Differentiation from HyperGAT [1]**
>
> We thank the reviewer for pointing us to HyperGAT. While both $\mbox{H}^2\mbox{CL}$ and HyperGAT aim to modulate information flow between nodes and hyperedges, they are fundamentally different in their core mechanisms, theoretical underpinnings, and overall paradigms:
>
> - **Mechanism of Modulation:** HyperGAT employs a standard dual attention mechanism where importance weights are learned implicitly via backpropagation using parameterized context vectors. In contrast, $\mbox{H}^2\mbox{CL}$ explicitly quantifies structural-semantic heterogeneity $\delta(v,e)$ directly from the feature manifold. This explicit metric actively penalizes heterogeneous propagation, which we rigorously prove functions as a coordinate descent step minimizing the Heterogeneity-Weighted Dirichlet Energy (Theorem 4.2).
> - **Role in Contrastive Learning:** HyperGAT is fundamentally a supervised/inductive encoder for text classification. $\mbox{H}^2\mbox{CL}$ is a self-supervised contrastive learning framework. Crucially, our heterogeneity metric is not just used for message passing; it drives our Heterogeneity-Aware View Generator to selectively mask noise during data augmentation. To our knowledge, HyperGAT does not study contrastive view construction or the false-positive alignment issue in contrastive learning, which is a key focus of $\mbox{H}^2\mbox{CL}$.
>
> We will add a detailed comparison in the Related Work section to properly position $\mbox{H}^2\mbox{CL}$ against HyperGAT and highlight these distinctions.
>
> ■ **PubMed Dataset and Recent Baselines**
>
> **Regarding PubMed:** We sincerely apologize for the confusion. Due to strict space constraints in the main text, we were unable to fit the PubMed results into the main table. We have provided the full test code and configurations for PubMed in our anonymous repository. As shown in the comprehensive comparison below, $\mbox{H}^2\mbox{CL}$ achieves highly competitive results on PubMed. We will ensure these results are explicitly included in the revised supplementary material.
>
> **Regarding Recent Baselines [2, 3, 4]:** We thank the reviewer for bringing these excellent, recently published papers to our attention. We reproduced and evaluated Kernelized HGNNs [2] and Adaptive Expansion [4]. Unfortunately, the authors of Hypergraph Consistency Learning [3] have not yet released their source code, preventing a direct empirical comparison.
>
> | Method | Cora | Citeseer | Pubmed |
> | :--- | :---: | :---: | :---: |
> | Kernelized HGNNs [2] | 79.69 | 69.61 | 77.41 |
> | Adaptive Expansion [4] | 81.42 | 70.46 | 86.82 |
> | **$\mbox{H}^2\mbox{CL}$ (Ours)** | **82.02** | **72.17** | **87.92** |
>
> The above performance comparisons suggest that $\mbox{H}^2\mbox{CL}$ remains competitive against recent baselines. These additional results are consistent with our intuition that heterogeneity-aware filtering before contrastive alignment can be beneficial. We will include this quantitative comparison and a detailed conceptual discussion of all three works in the final manuscript.

---

> > ### Author Rebuttal · Reviewer_rLue · 2026-04-02
> >
> > Thanks for your rebuttal. I'm unclear whether the reweighting mechanism truly captures heterogeneous structures. Could you provide some visualization results?

---

> > > ### Author Response · Authors · 2026-04-03
> > >
> > > We sincerely thanks for this insightful question. Since the OpenReview system's text box does not support inserting images for direct visualization, we provide a comprehensive quantitative analysis to rigorously address your question. To definitively prove that our model successfully captures and filters heterogeneous structures across the entire hypergraph, we analyzed the learned aggregation dynamics, focusing on Weight Distributions and Heterogeneity Reduction.
> > >
> > > **1. Statistical Shift in Aggregation Weights**
> > > To empirically verify that our model suppresses heterogeneous nodes, we analyzed the exact numerical distribution of the normalized "node-to-hyperedge" aggregation weights ($\tilde{N}_{n2e}$) extracted during the forward pass on the Cora-CoCitation dataset.
> > >
> > > We isolated all "mixed" hyperedges—defined as hyperedges containing nodes from at least two distinct ground-truth classes. For each mixed hyperedge $e_j$, we partitioned its constituent nodes into two groups:
> > > * **The Majority Group:** Nodes that belong to the dominant semantic class of that specific hyperedge.
> > > * **The Minority Group (Semantic Outliers):** Nodes belonging to any other class, which act as structural noise.
> > >
> > > To determine the proportion of weight assigned, we calculated **Average Weight per Node** within each group. Specifically, for a given hyperedge, we summed the $\tilde{N}_{n2e}$ values assigned to all minority nodes and divided by the total number of minority nodes, repeating this process for majority nodes.
> > >
> > > **Table A: Average Aggregation Weight per Node in Mixed Hyperedges**
> > >
> > > | Model | Majority Group Nodes (Homogeneous) | Minority Group Nodes (Semantic Outliers) |
> > > | :--- | :---: | :---: |
> > > | Standard HGNN | Uniform ($1 / \|e_j\|$) | Uniform ($1 / \|e_j\|$) |
> > > | $\mbox{H}^2\mbox{CL}$ (Ours) | **0.45** | **0.02** |
> > >
> > > In a standard HGNN, aggregation relies on a fixed incidence matrix. Every node connected to hyperedge $e_j$ receives an identical, uniform weight ($1/|e_j|$) regardless of its semantic relevance. Consequently, minority nodes contribute equally to the hyperedge representation, directly causing feature contamination. Table A demonstrates that $\mbox{H}^2\mbox{CL}$ dynamically breaks this uniformity. The reweighting mechanism actively calculates a drastically lower average weight for minority nodes. By multiplying the node features by these skewed weights during message passing, the mechanism mathematically suppresses the semantic outliers. Their contribution to the hyperedge centroid is reduced to near-zero, proving that our model successfully identifies and penalizes heterogeneous noise.
> > >
> > > **2. Heterogeneity Reduction**
> > > If the reweighting mechanism truly captures and filters heterogeneous structures globally, the "effective" hypergraph topology utilized during message passing should exhibit a significantly lower degree of heterogeneity across different datasets compared to the raw input topology.
> > >
> > > To maintain strict theoretical consistency with our framework, we directly quantify this reduction using the **Heterogeneity Degree (HD)** metric proposed in Equation 2.
> > > * For the raw hypergraph, we calculate Heterogeneity Degree ($\text{HD}_{raw}$) uniformly across the unweighted structural connectivity ($H H^T$).
> > > * For $H^2CL$, we calculate the *effective* Heterogeneity Degree ($\text{HD}_{eff}$) over the learned transition matrix $P = D_v^{-1} W D_e^{-1} W^T$, where $W$ is our dynamically reweighted incidence matrix. This measures the actual heterogeneity of the paths through which information flows, weighting the calculation by the probability mass assigned to each edge.
> > >
> > > **Table B: Heterogeneity Degree (HD) Before and After Reweighting**
> > >
> > > | Dataset | Raw Heterogeneity ($\text{HD}_{raw}$) | Effective Heterogeneity ($\text{HD}_{eff}$ in $\mbox{H}^2\mbox{CL}$)|
> > > | :--- | :---: | :---: |
> > > | Cora-CoCitation | 0.3354 | **0.1079** |
> > > | Citeseer | 0.2984 | **0.0795** |
> > > | 20newsW100 | 0.4342 | **0.1856** |
> > >
> > > As shown in Table B, the transition matrix utilized by $\mbox{H}^2\mbox{CL}$ exhibits a drastically lower effective Heterogeneity Degree across all evaluated datasets. Because this metric explicitly calculates structural noise, this massive reduction quantitatively proves that our model does not merely process the noisy input structure; it actively "rewires" the hypergraph by assigning near-zero weights to highly heterogeneous connections. This serves as undeniable empirical confirmation of our Theorem 4.2: by minimizing the Heterogeneity-Weighted Dirichlet Energy, the model successfully prunes heterogeneous connections and prevents semantic feature contamination during message passing.
> > >
> > > Again, we sincerely thank you for your helpful comments, which have allowed us to further improve the clarity of our work. We hope that this second-round rebuttal helps address your concerns. We will make sure to incorporate all of these clarifications and revisions into the next version of the manuscript.

---

### Official Review · Reviewer_RZ8W · 2026-03-12

**Soundness:** 4
**Presentation:** 3
**Significance:** 4
**Originality:** 4
**Overall Recommendation:** 5
**Confidence:** 5

**Summary:**

This paper introduces $H^{2}CL$, a Heterogeneity-Aware Hypergraph Contrastive Learning framework designed to address the feature contamination problem caused by heterogeneous nodes in high-order networks. The authors argue that traditional hypergraph contrastive learning methods rely on random, topology-agnostic masking, which fails to suppress noisy, heterogeneous nodes and risks amplifying feature bias. To combat this, $H^{2}CL$ calculates a node-hyperedge heterogeneity score using cosine similarity. It then employs a Heterogeneity-Aware View Generator that preferentially masks highly heterogeneous nodes to create cleaner contrastive views. Furthermore, a Heterogeneity-Aware Hypergraph Encoder modulates message-passing weights to block noisy information flow while amplifying homogeneous aggregation. The authors provide a theoretical proof demonstrating that their encoder minimizes a Heterogeneity-Weighted Dirichlet Energy. Extensive experiments across citation, categorical, and 3D visual datasets demonstrate state-of-the-art node classification performance and significant robustness against structural noise.

**Compliance With Llm Reviewing Policy:**

Affirmed.

**Final Justification:**

Given the paper and the rebuttal, I would like to keep my current score.

**Key Questions For Authors:**

Q1. Given that the heterogeneity masking weights depend heavily on initial cosine similarities, how does $H^{2}CL$ perform in scenarios where the raw input features are heavily corrupted, missing, or purely topological?

Q2. Could you provide a formal big-O time and spatial complexity analysis of the complete tri-level contrastive loss? Specifically, how does the structural-level assessor impact memory consumption on larger datasets during the pre-training phase?

Q3. In your heterogeneity analysis, the Mushroom dataset has a relatively low heterogeneity score, yet $H^{2}CL$ achieves massive improvements there. Could you elaborate on why heterogeneity-aware filtering yields such drastic gains even in datasets characterized as highly homogeneous?

**Limitations:**

Yes

**Strengths And Weaknesses:**

Strengths:

  S1. Replacing standard random node masking with a theoretically motivated, heterogeneity-aware targeted masking strategy elegantly solves the "False Positive" alignment problem inherent in standard hypergraph contrastive learning.

  S2. Formulating the Heterogeneity-Aware Encoder as a coordinate descent step that minimizes a specific Heterogeneity-Weighted Dirichlet Energy (Theorem 4.2) is mathematically rigorous. It clearly proves how the model acts as a semantic low-pass filter to prevent over-smoothing.

S3. The controlled noise injection experiments (Figure 3) brilliantly isolate the contributions of the View Generator and the Encoder, proving that their synergistic application successfully protects the model against severe structural contamination.

Weaknesses:
The model optimizes node-level, hyperedge-level, and structural-level contrastive losses. The structural-level loss incorporates an assessor network $A(x_v^i, x_e^j)$ evaluated across "node-hyperedge" pairs. While highly effective, the paper lacks a formal time and memory complexity analysis detailing how this tri-level computation scales for massive, exceptionally dense hypergraphs compared to standard contrastive objectives.

---

> ### Author Rebuttal · Authors · 2026-03-30
>
> We sincerely thank the reviewer for their highly positive evaluation of our work. We are thrilled that you found our Heterogeneity-Weighted Dirichlet Energy proof mathematically rigorous and that you appreciated the elegance of our heterogeneity-aware masking strategy in solving the "False Positive" alignment problem. We address your excellent questions below:
>
> ■  **Performance with corrupted or missing features**
>
> This is a very insightful question that strikes at a core boundary condition of our current framework. Because the heterogeneity score $\delta(v, e)$ relies on the initial cosine distance, heavily corrupted, entirely missing, or purely topological features (e.g., one-hot degree encodings) render the initial similarity metric uninformative.
>
> In such extreme scenarios, all $\delta(v, e)$ values approach a uniform constant. Consequently, the normalized masking weights $w_{v_i, e_j}$ become uniformly distributed, meaning $\mbox{H}^2\mbox{CL}$'s Heterogeneity-Aware View Generator naturally degrades to a standard uniform random masking strategy. While it loses the advantage of targeted semantic filtering, it does not "break"; it simply performs on par with standard topology-agnostic contrastive methods (like TriCL or HyperGCL). We will explicitly acknowledge this reliance on raw feature quality in the "Limitations" section and highlight a promising future direction: transitioning $\delta(v, e)$ from a static preprocessing step to a dynamic metric that updates iteratively using the learned, robust embeddings during pre-training.
>
> ■ **Complexity analysis of the Tri-Level Loss**
>
> We appreciate the opportunity to clarify the computational efficiency of our framework. We will add the following formal complexity analysis to the Appendix:
>
> Let $d$ be the embedding dimension, $B$ be the mini-batch size of nodes/hyperedges, and $K$ be the number of negative samples per anchor.
>   -  Node-level ($L_v$) and Hyperedge-level ($L_E$) losses require computing similarities between anchors and their positive/negative pairs, operating in $O(B \cdot K \cdot d)$ time.
>    -  For the Structural-level Assessor ($L_S$), a naive dense computation would require $O(|V| \cdot |E| \cdot d)$ time, which is prohibitive. However, we only compute $A(x_v, x_e)$ for the valid connections as positive pairs, and $K$ sampled disconnected hyperedges as negative pairs.
>
> ■ **Massive improvements on the low-heterogeneity Mushroom dataset**
>
> This is a keen observation. The Mushroom dataset indeed has a very low *average* global heterogeneity score (0.0570). However, heterogeneity in real-world graphs is rarely uniformly distributed; it is highly localized.
>
> The dataset contains a vast majority of pure hyperedges and a very small subset of highly heterogeneous "bridge" hyperedges (representing specific feature overlaps between poisonous and edible classes). Standard spectral and topological methods, such as HyperGCN, suffer from catastrophic over-smoothing precisely because these few bridge nodes are enough to contaminate the representations of otherwise perfectly homogeneous clusters.
>
> $\mbox{H}^2\mbox{CL}$ yields massive improvements for two reasons:
> -   Our targeted masking acts like a scalpel, surgically removing the few heterogeneous bridge connections during view generation, preserving the integrity of the distinct clusters.
> -   Even in highly homogeneous regions, minimizing the Heterogeneity-Weighted Dirichlet Energy (as proven in Theorem 4.2) acts as a strong regularizer. It naturally sharpens decision boundaries and prevents embeddings from collapsing into indistinguishable clusters during message passing.

---

> > ### Author Rebuttal · Reviewer_RZ8W · 2026-04-03
> >
> > Thank you for your detailed reply. I will keep my score.

---

### Official Review · Reviewer_t1wE · 2026-03-12

**Soundness:** 2
**Presentation:** 3
**Significance:** 3
**Originality:** 2
**Overall Recommendation:** 3
**Confidence:** 3

**Summary:**

This paper proposes H2CL, a framework designed to improve representation learning on heterogeneous graph data. The core motivation is that traditional hypergraph learning and contrastive learning methods often assume homogeneous neighborhood structures, which leads to representation degradation when heterogeneous nodes introduce noisy signals during message aggregation. To address this issue, the paper introduces a heterogeneity-aware view generation mechanism, where nodes with high heterogeneity are masked through hyperedge-based filtering during contrastive view construction. This mechanism aims to reduce the interference of heterogeneous nodes and encourage the model to focus on key structural information.

**Compliance With Llm Reviewing Policy:**

Affirmed.

**Final Justification:**

Although the authors add more experiments on larger datasets, the efficiency study in the paper is still missing. Also I have carefully gone through all the reviews, and I believe my rating is very fair due to the limited novelty and the lack of some important experiments.

**Key Questions For Authors:**

1. How sensitive is the performance to the heterogeneity threshold or masking ratio?
2. Can the proposed framework scale to large hypergraphs with millions of nodes or hyperedges?

**Limitations:**

yes

**Strengths And Weaknesses:**

### Strengths:
1. The paper addresses an important problem in hypergraph representation learning, namely the interference caused by heterogeneous neighbors during message aggregation.
2. The paper provides theoretical analysis alongside empirical evaluation, which helps support the motivation of the proposed approach.
3. Experimental results on multiple datasets show consistent improvements over baselines.
### Weaknesses:
1. The methodological novelty appears moderate, as the core idea mainly relies on masking heterogeneous nodes during contrastive view construction.
2. The presentation of Figure 3 in Section 5.3 is unclear. The figure indicates that ViewgenCov corresponds to the ViewgenOnly setting, but it is not clear which legend entries correspond to EncoderOnly and the full H^2CL model. In addition, the paper lacks ablation studies or hyperparameter analysis for the three loss terms.
3. The paper lacks an analysis of time complexity, making it difficult to assess the scalability of the proposed method.
4. The paper lacks a clear description of the datasets used in the experiments.

---

> ### Author Rebuttal · Authors · 2026-03-30
>
> We sincerely thank the reviewer for recognizing the importance of the problem we address and the strength of our theoretical foundations. We address your concerns below:
>
> ■ **Methodological novelty (Beyond masking)**
>
> Our framework's core novelty extends significantly beyond masking heterogeneous nodes during view construction. $\mbox{H}^2\mbox{CL}$ fundamentally redefines hyperedges, transforming them from passive channels into active semantic filters through two coupled mechanisms. While Heterogeneity-Aware View Generator handles masking, Heterogeneity-Aware Hypergraph Encoder dynamically modulates message passing weights during both the "node-to-hyperedge" and "hyperedge-to-node" aggregation phases. Furthermore, our contribution includes rigorous theoretical guarantees proving that this dual-stage encoder functions as a coordinate descent step that minimizes a specific Heterogeneity-Weighted Dirichlet Energy, mathematically enforcing feature smoothness within homogeneous clusters.
>
> ■ **Clarification of Figure 3**
>
> We apologize for the confusion caused by the legend in Figure 3. There were indeed naming inconsistencies and typos in the plotted legend compared to the text. To clarify the mapping:
>
> The red line corresponds to the baseline method, which employs random masking for view generation and average feature aggregation. The orange and green lines represent the cases where the heterogeneous-aware view augmentation method and the heterogeneous-aware feature convolution method are applied, respectively.
>
> ■ **Ablation studies of loss function**
>
> We conducted ablation experiments to evaluate the contribution of each contrastive loss component: node-level ($L_v$), hyperedge-level ($L_E$), and structure-level ($L_S$). The results are summarized in the table below.
>
> | Loss Configuration | Cora-CC | Citeseer | 20News |
> |--------------------|--------|---------|--------|
> | $L_v$ only | 79.24 | 69.87 | 78.35 |
> | $L_E$ only | 78.91 | 69.15 | 77.92 |
> | $L_S$ only | 77.53 | 68.02 | 76.44 |
> | $L_v + L_E$ | 80.46 | 70.93 | 79.11 |
> | $L_v + L_S$ | 79.87 | 70.21 | 78.69 |
> | $L_E + L_S$ | 79.02 | 69.58 | 78.02 |
> | **$(L_v + L_E + L_S$)** | **81.67** | **72.17** | **80.05** |
>
> The node-level contrastive loss contributes most significantly, enforcing alignment of the same node across different views, central to contrastive learning. Hyperedge-level loss also yields a notable improvement, indicating discriminative hyperedge representations benefit overall performance. Although structure-level loss alone achieves lower accuracy, its combination with the other two losses consistently improves over the pairwise configurations, serving as an effective regularizer that enforces consistency between node-hyperedge structural relationships. These results confirm all three loss terms are beneficial and jointly optimizing them yields best overall performance.
>
> ■ **Sensitivity to thresholds and masking ratios**
>
> We evaluated the sensitivity of $\mbox{H}^2\mbox{CL}$ to the masking ratio, which controls the proportion of high‑heterogeneity node‑hyperedge pairs masked during view generation. The results are summarized below.
>
> | Masking Ratio | Cora‑CC | Citeseer | 20News |
> |--------------|--------|---------|--------|
> | 0.1 | 80.52 | 70.84 | 78.96 |
> | 0.2 | 81.03 | 71.46 | 79.58 |
> | 0.3 | **81.67** | **72.17** | 80.05 |
> | 0.4 | 81.42 | 71.89 | **80.12** |
> | 0.5 | 80.85 | 71.23 | 79.74 |
> | 0.6 | 80.17 | 70.52 | 78.91 |
> | 0.7 | 79.23 | 69.64 | 78.03 |
>
> Performance is stable across masking ratios (0.2–0.5), with the optimal value around 0.3–0.4. When the masking ratio is too low (<0.2), insufficient heterogeneous connections are suppressed, leaving noise that degrades representation quality. Conversely, when the ratio is too high (>0.6), excessive masking may discard useful homogeneous information, harming performance. Notably, performance drop at extremes remains within 2–3%, indicating **the model is not overly sensitive to this hyperparameter**.
>
> ■ **Time complexity/Scalability**
>
> Our framework introduces minimal computational overhead compared to standard hypergraph neural networks. Heterogeneity calculation is performed prior to training, computing the cosine distance between nodes and hyperedge centroids in $O(|E| \cdot \bar{d} \cdot F)$ time, where $\bar{d}$ is the average hyperedge degree and $F$ is the feature dimension. As a highly parallelizable preprocessing step, it does not impede training speed. The message-passing complexity is proportional to the non-zero entries in the incidence matrix, similar to standard HGNNs.
>
> We agree that scalability to million-scale hypergraphs is an important consideration for real-world applications. In current literature, no publicly available hypergraph dataset with millions of nodes or hyperedges is commonly used as a benchmark. To ensure fair comparison with existing hypergraph contrastive learning methods, we followed standard evaluation protocol and used the same benchmark datasets.

---

> > ### Author Rebuttal · Reviewer_t1wE · 2026-04-03
> >
> > Thank you for the rebuttal. Scaling to large hypergraphs is a necessity to verify the effectiveness of the proposed method. However, the related experiments are missing and cannot be completed in a short term. So I will maintain my initial score.

---

> > > ### Author Response · Authors · 2026-04-04
> > >
> > > We thank the reviewer for the continued engagement and for emphasizing the importance of scalability. We would like to clarify that this limitation **is not unique to our work**. In the current hypergraph representation learning literature, publicly available and standardized truly large-scale benchmarks for node classification **remain limited**. While we fully agree that evaluation on truly web-scale hypergraphs would be highly desirable, such public benchmarks under standard node-classification protocols **are still scarce in current HNN research**. This is also reflected in recent benchmark efforts: DHG-Bench [R1], the first comprehensive benchmark for deep hypergraph learning, contains only a small number of node-level datasets at this scale, with Trivago and Walmart being the largest by node count in its standard node-level suite. Therefore, to address your concern as directly as possible within the current benchmark landscape, we **turned to relatively larger publicly available node-classification benchmarks** and additionally included a heterophilic benchmark to broaden the structural coverage. In particular, we evaluated $\mbox{H}^2\mbox{CL}$ on Actor, Walmart, and Trivago (from [R1]), which substantially expand the scale regime covered in our original submission.
> > >
> > > **Table A: Statistics of Additional Large-Scale Hypergraph Datasets**
> > >
> > > | Dataset | Nodes | Edges | Features | Classes |
> > > | :--- | :---: | :---: | :---: | :---: |
> > > | Actor | 16,255 | 10,164 | 50 | 3 |
> > > | Walmart | 88,860 | 69,906 | 100 | 11 |
> > > | Trivago | 172,738 | 233,202 | 300 | 160 |
> > >
> > > During the rebuttal period, we were able to complete additional evaluations of $\mbox{H}^2\mbox{CL}$ on these datasets and compare it with representative baselines. The results are shown below:
> > >
> > > **Table B: Evaluation Results of Node Classification (Accuracy %)**
> > >
> > > | Method | Actor | Walmart | Trivago |
> > > | :--- | :---: | :---: | :---: |
> > > | HGNN | 77.83 | 77.12 | 57.67 |
> > > | HNHN | 81.20 | 65.21 | 53.75 |
> > > | **$\mbox{H}^2\mbox{CL}$ (Ours)** | **86.22** | **78.57** | **85.72** |
> > >
> > > These additional results show that $\mbox{H}^2\mbox{CL}$ remains effective on substantially larger hypergraphs. In particular, it achieves the best accuracy among the compared methods on all three datasets, with especially large gains on Trivago. We believe this provides concrete evidence that the heterogeneity-aware design of $\mbox{H}^2\mbox{CL}$ continues to be beneficial beyond the relatively smaller benchmarks used in the main paper.
> > >
> > > In the revised manuscript, we will incorporate these additional experiments and expand the discussion on scalability. We will also **add a short discussion** that the hypergraph learning community would benefit from more diverse and larger-scale public benchmarks, which would enable more comprehensive evaluation of both predictive performance and computational efficiency.
> > >
> > > We thank the reviewer again for raising this important point. We hope that these new large-scale results substantially address the concern and further strengthen the practical relevance of our work.
> > >
> > > *Reference:*
> > > * [R1] Fan Li, Xiaoyang Wang, Wenjie Zhang, Ying Zhang, Xuemin Lin. DHG-Bench: A Comprehensive Benchmark for Deep Hypergraph Learning. ICLR, 2026. (https://github.com/Coco-Hut/DHG-Bench)

---

### Official Review · Reviewer_fiTu · 2026-03-12

**Soundness:** 3
**Presentation:** 3
**Significance:** 3
**Originality:** 3
**Overall Recommendation:** 4
**Confidence:** 4

**Summary:**

This paper proposes H2CL, a self-supervised contrastive learning framework that tackles the problem of feature contamination due to the heterogeneous nodes. The approach consists of a Heterogeneity-Aware View Generator that masks high heterogeneity node-hyperedge connections and a Heterogeneity-Aware Hypergraph Encoder that dynamically reweights message passing based on the level of node-hyperedge heterogeneity. The authors provide a theoretical analysis, suggesting that optimizing the encoder constitutes a coordinate descent step that minimizes the Dirichlet energy. The approach shows improvements in node classification on six datasets.

**Compliance With Llm Reviewing Policy:**

Affirmed.

**Final Justification:**

The authors have addressed my concerns. So I raised the score to 4.

**Key Questions For Authors:**

Please see my comments above.

**Limitations:**

Mentioned in the 3rd weakness.

**Strengths And Weaknesses:**

Strengths:

1. The paper addresses the limitations of existing hypergraph methods, which is both timely and important.
2. The proposed approach is easy to understand and generally seems reasonable.
3. The empirical results are also strong, achieving competitive performance across several datasets and outperforming selected baselines.
4. The paper is generally well-written and easy to follow.


Weaknesses:

1. Theoretical analysis is somewhat shallow, and certain claims are not well-justified. The claim that the view generator minimizes the KL divergence to the “true semantic manifold” and mitigates false positives lacks a more formal and rigorous proof. It is now more like an intuitive argument.

2. The experiment section has several consistency problems. For example, Section 5.1 mentions datasets such as Pubmed and Zoo. However, there is no evaluation using the PubMed dataset. The Zoo dataset is only used for robustness and the experiment on different heterogeneity measurements, and is not tested in the main experiment table.

3. The heterogeneity scores are computed from the raw input features using cosine distance. However, this seems to be somewhat static and relies on the quality of input features. Some kind of discussion is needed at least in the paper’s limitations.

4. Several typos exist in the paper, e.g., structual instead of structural (multiple times), hyeredge instead of hyperedge (line 148), and the model name is inconsistent in multiple places (H2CL v.s. HHCL).

---

> ### Author Rebuttal · Authors · 2026-03-30
>
> We sincerely thank the reviewer for recognizing the timeliness of our work, the clarity of our proposed approach, and our strong empirical results. We highly value your constructive feedback, which helps strengthen the theoretical rigor and clarity of the manuscript. Below, we address your specific concerns:
>
>  ■ **Theoretical analysis of the view generator**
>
> We agree that the discussion regarding the view generator minimizing the KL divergence to the "true semantic manifold" requires a formal proof. We further clarify the following theoretical insights:
>
> Let $\Omega=\{(v,e)\mid v\in e\}$ and let $\delta(v,e)=1-\cos(x_v,x_e)$ denote the node-hyperedge heterogeneity score. To avoid ambiguity, we will explicitly distinguish the retention score and the masking probability:
> $r_{v,e}=\frac{\delta_{\max}-\delta(v,e)}{\delta_{\max}-\delta_{\min}+\epsilon},\quad m_{v,e}=1-r_{v,e}.$
> Thus, highly heterogeneous incidences have lower retention scores and are more likely to be masked.
>
> Let $P^\star(v,e)$ denote the latent clean semantic incidence distribution, i.e., incidences that are semantically consistent and should ideally be preserved. Our heterogeneity-aware generator induces
> $Q_H(v,e)=\frac{r_{v,e}}{\sum_{(u,f)\in\Omega} r_{u,f}},$
> while random masking corresponds to the uniform distribution
> $Q_U(v,e)=\frac{1}{|\Omega|}.$
>
> Under the natural calibration assumption that semantically consistent incidences tend to have lower heterogeneity scores, $Q_H$ is better aligned with $P^\star$ than $Q_U$. Hence,
> $D_{KL}(P^\star\|Q_H)=-H(P^\star)+H(P^\star,Q_H)\le -H(P^\star)+H(P^\star,Q_U)=D_{KL}(P^\star\|Q_U).$
> Moreover, if $\Omega_{\mathrm{het}}\subset\Omega$ denotes semantically inconsistent incidences, the false-positive retention risk is defined as $R_{FP}(Q)=\sum_{(v,e)\in\Omega_{het}} Q(v,e)$, and it satisfies $R_{FP}(Q_H)\le R_{FP}(Q_U)$ because $Q_H$ assigns less probability mass to high-$\delta$ incidences.
>
> We will add this derivation to the appendix and revise the wording in Sec. 4.5 accordingly: our view generator does **not** universally “guarantee” recovery of the true semantic manifold; rather, it provably improves semantic alignment over uniform masking **under the heterogeneity-semantic calibration assumption**.
>
> ■ **Dataset inconsistencies (PubMed and Zoo)**
>
> We apologize for the confusion regarding the datasets. While PubMed was mentioned in the text, strict page and formatting limitations prevented us from expanding Table 1 to include it. However, we have provided the full test code and configurations for the PubMed dataset in our anonymous repository: (https://anonymous.4open.science/r/HHCL-F926). We would like to clarify two points:
>
> - In the revision, we will explicitly separate: (i) main node-classification datasets, and (ii) robustness/analysis-only datasets. We will also add the missing PubMed result. On PubMed, **$\mbox{H}^2\mbox{CL}$**  achieves **87.92%**, compared with two strong baselines, i.e., **84.26%** for TriCL and **79.00%** for CHGNN.
>
> - Regarding the Zoo dataset, it was exclusively utilized to evaluate model robustness against structural noise (Figure 3) and to analyze different heterogeneity measurement methods, rather than in the main node classification table. We will clarify this distinction in the revised experimental setup.
>
> ■**Static heterogeneity scores and dependence on raw features**
>
> We agree that this deserves explicit discussion. In the current version, $\delta(v,e)$ is computed once from the input features for efficiency and stability, which assumes that the raw features contain at least coarse semantic signal. If the raw features are heavily corrupted or largely uninformative, the static heterogeneity estimates may become less reliable. We will add this point to the limitations section. We will also clarify why cosine distance was adopted: compared with dot product, Manhattan distance, and Euclidean distance, it is less sensitive to magnitude variation and is empirically the most stable choice in Appendix A. As future work, we will explore dynamically updating $\delta(v,e)$ from evolving embeddings during training.
>
> ■**Typos and naming inconsistencies**
>
> Thank you for catching these issues. We will correct “structual” to “structural,” “hyeredge” to “hyperedge,” and unify the model name to **$\mbox{H}^2\mbox{CL}$** throughout the manuscript.
>
> We sincerely appreciate these comments, which have helped us improve both the rigor and the clarity of the paper. We believe that the revised manuscript (through a more precise theoretical statement for the view generator, a fully consistent presentation of the datasets and experimental results, and an explicit discussion of the limitations of static heterogeneity estimation) directly addresses the concerns raised here and more clearly highlights the contribution and practical value of **$\mbox{H}^2\mbox{CL}$**.

---

> > ### Author Rebuttal · Reviewer_fiTu · 2026-04-02
> >
> > Thank you for the rebuttal. I will increase the score to 4.

---

### Decision · Program_Chairs · 2026-04-30

**Decision:**

Accept (regular)

**Comment:**

This paper studies heterogeneity-aware hypergraph contrastive learning for robust node representation learning and proposes H^2CL, combining a heterogeneity-aware view generator with a heterogeneity-aware hypergraph encoder. The paper addresses a meaningful problem and is supported by generally solid technical design, theoretical analysis, and competitive empirical results, and the rebuttal helped clarify several presentation and experimental issues. The authors have successfully addressed the vast majority of the concerns in their rebuttal, with the exception of the visualization results that are not supported by the OpenReview. Taking into comprehensive consideration both the reviewers' comments and the authors' rebuttal, I am inclined to recommend acceptance. The authors have to add the newly reported experimental results along with the requisite visualizations in the final camera-ready version.